# Who falls between the cracks? Identifying eligible PrEP users among people with Sub-Saharan African migration background living in Antwerp, Belgium

**Veerle Buffel**[1]*, **Caroline Masquillier**[1], **Thijs Reyniers**[2], **Ella Van Landeghem**[2], **Edwin Wouters**[1], **Bea Vuylsteke**[2], **Christiana Nöstlinger**[2]

1 Department of Sociology, University of Antwerp, Antwerp, Belgium, 2 Department of Public Health, Institute of Tropical Medicine, Antwerp, Belgium

* veerle.buffel@uantwerpen.be

## Abstract

### Introduction

This study produces an estimate of the proportion of eligible PrEP users among people of Sub-Saharan African background based on the Belgian PrEP eligibility criteria and examines associations with socio-economic and demographic characteristics.

### Methods

We performed logistic regression analysis on data of a representative community-based survey conducted among Sub-Saharan African communities (n = 685) living in Antwerp.

### Results

Almost a third (30.3%) of the respondents were eligible to use PrEP. Those who were male, single, lower educated, undocumented, and had experienced forced sex were more likely to be eligible for PrEP use. The findings highlight the importance of taking intra-, interpersonal and structural HIV risk factors into account.

### Conclusions

The study shows high unmet PrEP needs in this population, especially among those with high vulnerability for HIV acquisition. A better understanding of barriers to PrEP use in this population group is needed to allow for equitable access.

## Introduction

People with a Sub-Saharan African (SSA) migration background are the second largest group affected by HIV in Belgium [1]. In 2019, a total of 923 new HIV diagnosis were identified in

**Data Availability Statement:** All relevant data are within the paper and its Supporting information files.

**Funding:** This study is part of the Promise project 'Optimise PrEP to Maximise Impact', funded by FWO-SBO (Flemish Scientific Research – Strategic Basic Research) to BV (S004919N)".

**Competing interests:** The authors have declared that no competing interests exist.

Belgium, among them 349 were through heterosexual transmission (51% of the cases with known transmission type) [2]. People with SSA migration background constituted 48% of all new HIV cases with a heterosexual transmission mode in 2019, 67% among them were women [2]. An HIV prevalence study among SSA communities in Antwerp found a prevalence of 5.9% among women and 4.2% among men [1]. About one in three people in this population was estimated to have acquired HIV post-migration in European host countries [3–5]. This supports the notion that primary HIV prevention should be strengthened in this group.

Several countries, including Belgium, have taken up Pre-exposure Prophylaxis (PrEP) into their public health response to HIV by reimbursing it for people at high risk of HIV acquisition [6, 7]. Most countries have developed specific criteria to regulate access to PrEP. In Belgium, national eligibility criteria provide both specific criteria targeting men who have sex with men (MSM) and general criteria for people at high-risk of HIV acquisition (Fig 1). However, PrEP uptake has not been equal across these two groups and currently mainly MSM are using PrEP: in 2019, 98% of the PrEP starters in Belgium were MSM and 77% had a Belgian nationality [2].

The high HIV prevalence among SSA communities found in Antwerp suggests that many people belonging to this group could meet these eligibility criteria and thus benefit from PrEP [8]. However, in 2019 only 1.4% of all applications for PrEP reimbursement stemmed from people with SSA background (equaling to 25 people out of 1837 PrEP applicants with a known nationality) of whom 18 were MSM [2]. Research in neighboring countries, such as France, also found barriers to PrEP use among Sub-Saharan immigrants [9, 10].

We currently do not know what proportion of SSA communities may be eligible for PrEP, because they have been largely overlooked in PrEP research [11]. The absence of specific eligibility criteria may prevent healthcare providers from targeting those at risk of HIV acquisition for PrEP or PrEP referral [12]. Research has shown that ethnic minority groups at risk of HIV acquisition may not address their prevention needs spontaneously with healthcare providers due to social desirability, discomfort with sharing sensitive information and fear of provider judgement or stigma [12]. In addition, HIV prevention demand has found to be generally low among SSA communities and other factors such as insufficient knowledge and misconceptions about PrEP may prevent them from accessing PrEP [13, 14].

**Risk factors allowing for reimbursement**
I. MSM (men who have sex with men):
• who have had condomless anal sex with at least 2 partners in the last 6 months
• who have had multiple Sexually Transmitted Infections (STIs) (Syphilis, Chlamydia, Gonococcus or a primary infection with hepatitis B or C) during the last year
• who needed Post exposure Prophylaxis (PEP) several times during the last year
• who use psychoactive substances during sexual activities.
II. High-risk persons:
• People who inject drugs (PWID) who share needles
• People in sex work who are exposed to condomless sex
• People in general who are exposed to condomless sex with a high risk of HIV infection
•Partners of people living with HIV (PLWH) without viral suppression (recently started on treatment or no viral suppression with adequate treatment)

**Fig 1. Box 1: Eligibility criteria for PrEP reimbursement in Belgium (issued by the National institute for Health and Disability Insurance as of June 1, 2017).** Source: Rijksinstituut voor ziekte – en invaliditeitsverzekering (http://www.inami.fgov.be/nl/themas/kost-terugbetaling/door-ziekenfonds/geneesmiddel-gezondheidsproduct/terugbetalen/specialiteiten/wijzigingen/Paginas/geneesmiddelen-PrEp-HIV.aspx#.WqErna0zU3E).

Against this background, the study objectives are twofold: First, we aim to identify the proportion of SSA migrants who are theoretically eligible for PrEP use, based on the current eligibility and reimbursement criteria in Belgium; second, we examine the socio-economic and demographic characteristics which are related to these criteria among SSA migrants. This evidence is relevant to guide healthcare providers in accurately assessing HIV risk and potential PrEP needs among people of SSA origin.

## Materials and methods

### TOGETHER study

We performed a secondary analysis of data from a representative community-based bio-behavioral cross-sectional survey (i.e. TOGETHER project) to assess HIV prevalence conducted in 2013–2014 in Antwerp. Ethical approval for the TOGETHER project was obtained from the Institutional Review Board of the Institute of Tropical Medicine and the ethical committee of the University Hospital Antwerp. To be eligible, potential study participants had to be willing and able to provide written informed consent. More information can be find in the protocol paper of the TOGETHER project [1].

The TOGETHER data are the most recent behavioral data available for this sub-population in Belgium. The study used a two-stage time-location sampling (TLS) to obtain a venue-based sample of n = 744 adult Sub-Saharan African migrants in Antwerp [1]. A TLS takes advantage of the fact that some hard-to-reach populations tend to gather at certain types of sites/clusters at certain times. A list of these sites was prepared in a formative study and formed the sampling frame: at the first level, clusters (or sites) were randomly selected with a probability proportional-to-size and at the second level, a fixed number of study participants were randomly selected from each cluster.

All individuals socializing in a given setting at the time of the study visit (available attendance data) were eligible for inclusion in the survey if they met the following criteria: (1) self-identified sub-Saharan African migrant; (2) age 18 years or above; (3) accepting to answer the questionnaire; (4) accepting to provide an oral fluid sample; and (5) providing written informed consent. Recruitment, data collection and weighting procedures to adjust for unequal selection probability are described elsewhere [1, 15]. For the current analysis, HIV positive participants were excluded (32 HIV positive out of 717 respondents with valid information about their HIV status), since they would not be eligible for PrEP [7]–resulting in a sample size of 685 inidviduals.

### Operationalization of the eligibility criteria

The main variables of interest are variables corresponding to the Belgian PrEP eligibility criteria (Fig 1). As specific criteria for people with SSA migrant background are lacking, we adapted both the MSM-specific and general criteria to the target population of this study (Fig 2). As not all Belgian PrEP eligibility criteria could be measured directly, proxies were used approaching the original criteria as much as possible. These were developed based on existing scientific evidence and expert advice [1, 16, 17]. Box 2 (Fig 2) shows the eligibility guidelines and their operationalization based on the available information in the TOGETHER survey. In what is to follow, we discuss the criteria for which proxies were used.

For the criterion *'People in sex-work and exposed to condomless sex'* (eligibility criterion 6) we used the question about transactional sex (i.e. sex in exchange for gifts, food, money, papers or housing). In particular, young women living in and/or from SSA are known to engage in transactional sex as a way to make ends meet. It has been established as an important HIV risk factor among SSA migrants during and after settlement [16] and was found to be related to

| | Indicator (national criteria) | Variables in the TOGETHER data | Operationalized proxy |
|---|---|---|---|
| 1 | **MSM who have had condomless anal sex with at least 2 partners in the last 6 months** | - In the last 12 months, how many different stable and casual partners did you have sex with? (number)<br>- Did you use a condom the last time you had sex? (Yes/no)<br>*- No information about anal sex* | Having sex with at least four different partners during the last 12 months and the last time was without condom[1] |
| 2 | **MSM who have had multiple Sexually Transmitted Infections (STI) (Syphilis, Chlamydia, Gonococcus or a primary infection with hepatitis B or C) during the last year** | - Been diagnosed with a STI other than HIV (chlamydia, gonorrhea, syphilis,..) (yes/no)<br>- When was the last time you have been diagnosed with a STI? (<6 months ago/ between 6-12 months/ 1-2 year/ 2-5 year/ > 5 years) | Diagnosed with a STI less than 6 months ago[2] |
| 3 | **MSM who needed Post Exposure Prophylaxis (PEP) several times a year** | *- No information available about PEP-use* | No proxy available |
| 4 | **MSM who use psychoactive substances during sexual activities** | - Did you use alcohol and/or drugs (marijuana/hashish or other drugs) the last time you had sex with this partner? (yes/no) | Having used alcohol and/or drugs during sex the last time and not using a condom |
| 5 | **People who inject drug (PWID) who share needles** | *- No information available about injectable drug use* | No proxy available |
| 6 | **People in sex work who are exposed to condomless sex** | - In the last 12 months did you have sex with someone in exchange for gifts, food, money, papers or housing?<br>- Did you use a condom the last time you had sex? | Having condomless transactional sex in the last 12 months |
| 7 | **People in general who are exposed to condomless sex with a high risk of HIV infection** | **7.1 Concurrent relationship and low likelihood of future condom use**<br>- In last 12 months did you have a partner of whom you think he/she had other sexual partner(s) besides you?<br>- Did you use a condom the last time you had sex?<br>- How likely is it that you will use a condom with a new sexual partner in the future? | Being in a concurrent relationship in the last year or thinking that your partner is, having condomless sex and low likelihood of future condom use |
| | | **7.2 Sex with a member of a high-risk group and unaware of his/her HIV status**<br>- What is the country of origin of the last person you had sex with? (African/ Belgian/ Other)<br>- Do you know your last sexual partner's HIV status?<br>- Did you use a condom the last time you had sex? | Having condomless sex with someone of sub-Saharan African origin and being unaware of his/her HIV status |
| | | **7.3 Mobility and sexual behavior**<br>- After migrating, did you ever travel back to Africa and/or to another European country than Belgium? (Yes/No)<br>- When you travelled to Africa/Europe did you have sex there?<br>- When was the last time this happened?<br>- Did you use a condom?<br>- What type of partner was he/she (casual and living in Africa/another European country than Belgium; casual and travelling with; regular and living in Africa/another European country than Belgium; regular and travelling with) | Being sexually active on African/European travel(s) with a casual partner or a regular partner living in Africa/another European country than Belgium, without using a condom (in the last 12 months) |
| 8. | **Partners of people living with HIV (PLWH) without viral suppression[3]** | *- No information available about viral suppression* | No proxy available |

**Fig 2. Box 2: The MSM specific and high-risk group eligibility criteria adjusted to SSA migrants and operationalized by the TOGETHER data.** Notes: [1]We have doubled the number of sexual partners, as the period is doubled (the number of sexual partners was asked for the last 12 months instead of 6 months) and we do not have information about anal sex. [2] The guidelines consider the number of STIs during last year, while in the TOGETHER study participants only reported about lifetime STI and time of occurrence. Therefore, the time period from 12 months

was restricted to the last 6 months to operationalize a similar conservative measure. [3] Partners new on treatment or no viral suppression with adequate treatment.

social and economic hardships (such as having an undocumented status) [16], which in turn are HIV vulnerability factors [18].

To operationalize eligibility criterion 7 *'People in general who are exposed to condomless sex with a high risk of HIV infection'* we rely on recent publications focusing on important risk factors among SSA migrants [1, 16, 17]. Three SSA migrant-specific factors (not yet covered by one of the other national criteria) were selected and they are counted as separate criteria. The first two risk factors (item 7.1 and 7.2) consider sexual concurrency among SSA migrants and assortative sexual mixing [19], which is relevant in concentrated epidemics for HIV transmission. Preferences for African sexual partners, as well as concurrency within sexual networks where both high HIV prevalence or high rates of undiagnosed HIV elevating the risk of HIV transmission were established based upon the literature [1, 5, 18, 20]. The first factor is operationalized as 'being in a concurrent partnership, having condomless sex and a low likelihood of future condom use (item 7.1). A social desirability bias when reporting condom use may occur, so PrEP should be considered for people reporting any intercourse without a condom or concerns about their future use of condoms [7]. The second risk factor is operationalized as 'having condomless sex with an African partner of unknown HIV status' (item 7.2).

The third specific risk factor of interest to our target population is related to migrants' condomless sex during travelling abroad (eligibility criteria 7.3). Research showed that migrants' mobility was associated with increased risk for HIV infection [21] and that people of SSA origin traveled frequently both within Europe and to African home countries [1]. Studies in Amsterdam [22], London [23], Antwerp and Lisbon [17] support the relation between migrants' mobility after settling and increased HIV risk. The latter study showed that SSA migrants who travel–in Europe or to Africa–are at increased risk for HIV, reporting more condomless sex and concurrency than non-travelers [17]. Therefore, we combined the following two items: 'travelled to an African country' and 'travelled to a European country'. The third criteria is thus measured as 'having condomless sex on African/European travel(s) with a casual or regular partner living in Africa/another European country than Belgium'.

## Study variables

**Dependent variable.** The dichotomous variable 'being eligible to use PrEP (yes = 1; no = 0)' was operationalized as meeting at least one of the seven criteria–as presented in Box 2 (Fig 2). Study subjects could have a missing value for maximum 3 out of 7 items.

**Independent variables.** Socio-demographic variables included gender, age, sexual orientation, relationship status, country of origin, and migration duration [3, 5, 18, 20, 24]. *Gender* was a dichotomous variable (man/woman) and *age* was categorized in three age groups '18 to 30 years old', '31–40' and '41 or older'. *Relationship status* was combined with cohabitating with a partner or not (single, in a relation and cohabiting, and in a relation and not cohabiting with this partner). For *country of origin* we categorized the countries in regions (Central, Western, Southern and Eastern), and Southern and Eastern Africa were taken together because of small percentages. *Migration duration* (or migration history) consisted of four categories: 1) not living in Belgium; 2) recently migrated (living in Belgium since less than 2 years); 3) living in Belgium between 2 and 10 years; and 4) for more than 10 years or born in Belgium (second generation migrant). The variable *MSM* (yes/no) was constructed from the question 'in general, are the people you have sex with men, women or both?' whereby men who indicated to have sex with men or both men and women were labelled as MSM for our analysis.

In addition, socio-economic vulnerability was considered an independent variable, as previous research revealed a positive relation between socio-economic hardship and HIV-risk behavior among migrants [24]. Low or no educational level, unemployed or non-employed (retired, student, inactive due to disability), unstable housing, financial problems and being undocumented were considered as proxies for socio-economic vulnerability. They were measured by the categorical variable *educational level* (primary school or less, completed secondary, continued education); *employment status* ((self-) employed, unemployed, full time student, and non-employed); *financial problems* (no versus sometimes/most of the time); *unstable housing* (yes/ no) and *undocumented* (yes/no). The latter was based on the question: 'Do you currently have health insurance?' and not having any kind of health insurance (Belgian welfare system, health coverage via asylum centre, health insurance in another European country or in an African country') was considered a proxy of being undocumented [1]. Unstable housing included those who were homeless or living temporarily with friends. The variable *forced sex* was considered a risk factor for HIV infection and measured by the question 'forced sex' (lifetime) [4].

The study sites where study participants were selected were categorized in five types of settings and this is included as control variable: bars/parties of African organization, churches, public place (park, street, square), events and meetings of African organizations, and other (e.g. shop, hair salon, library, asylum center) [1].

## Analytic strategy

In a first step, we examined potential associations between socio-demographic and -economic variables and the eligibility criteria (separately, combined with and without the SSA migrant-specific criteria) using bivariate statistics resulting in a contingency table and bivariate logistic regressions (Table 1 and S2 Table). Wald Chi-square tests were used to determine whether the associations between these categorical variables were significant and the strength of the associations were measured by unadjusted (or crude) odds ratio's (OR). For all analyses, weighted data were used accounting for the unequal probability of selection according to Karon & Wejnert (2012) [25] (venue attendance, study participation, and sampling fraction [1]). SSA migrants who visited sites more frequently had a higher probability of selection in the study. Adjustment for this unequal selection probability was completed by calculating individual weights, based on the attendance [14].

Next, we performed multivariable logistic regression analyses with weighted data to investigate which factors were associated with 'being eligible to use PrEP', while controlling for confounding factors. Strengths of associations were measured using adjusted ORs and Wald Chi-square tests were applied to determine whether the individual coefficients of the regression were significant (with a p value < 0.05 considered statistically significant). We employed logistic regression models using a stepwise approach: to socio-demographic and migration related factors (Model 1) we added socio-economic factors (Model 2) and in the last step we also included the variable *forced sex* (Model 3). To get a better understanding of each HIV risk factor, the same analyses were also done with the individual eligibility criteria as dependent dichotomous variables. These results and their interpretation can be found respectively in S2 Table and S1 File.

All the analyses were performed by the statistical software IBM SPSS statistics 26 and the minimal anonymized data set is available as (S2 File).

## Results

Applying the exclusion criteria for this secondary analysis as described above resulted in a study sample of n = 685 participants. About 3.2% (n = 22) subjects had more than three

**Table 1. Factors associated with eligibility to PrEP in bivariate and multivariable logistic analysis (weighted data).**

| Variable (N total)[a] | Total N | % | % eligible | Un-adjusted OR | 95%-CI | | Adjusted OR | 95%-CI | |
|---|---|---|---|---|---|---|---|---|---|
| **Age** | 663 | | | | | | | | |
| Between 18–30 years old (ref.)[b] | 240 | 36.17 | 30.13 | 1.00 | | | 1.00 | | |
| Between 31–40 years old | 249 | 37.54 | 29.84 | 0.99 | 0.67 | 1.46 | 1.30 | 0.80 | 2.10 |
| Older than 41 years old | 174 | 26.29 | 31.03 | 1.05 | 0.69 | 1.60 | 1.50 | 0.85 | 2.65 |
| **Gender** | 663 | | | | | | | | |
| Women (ref.) | 239 | 36.01 | 25.52 | 1.00 | | | 1.00 | | |
| Men | 424 | 63.99 | 33.02 | **1.43** | **1.01** | **2.04**\* | **1.83** | **1.18** | **2.82**\*\* |
| **MSM** | 663 | | | | | | | | |
| No MSM (ref.) | 650 | 98.06 | 29.69 | 1.00 | | | 1.00 | | |
| MSM | 13 | 1.94 | 61.54 | **3.66** | **1.18** | **11.37**\* | 2.69 | 0.76 | 9.59 |
| **Relation status** | 662 | | | | | | | | |
| Not in a relationship (ref.) | 257 | 38.81 | 34.24 | 1.00 | | | 1.00 | | |
| In a relation and cohabiting | 285 | 42.97 | 24.21 | **0.61** | **0.42** | **0.89**\*\* | **0.58** | **0.36** | **0.92**\* |
| In a relation and not cohabiting | 120 | 18.22 | 36.67 | 1.11 | 0.71 | 1.71 | 1.09 | 0.65 | 1.83 |
| **Region of Origin** | 663 | | | | | | | | |
| Western Africa (ref.) | 443 | 66.86 | 28.89 | 1.00 | | | 1.00 | | |
| Central Africa | 180 | 27.22 | 30.94 | 1.10 | 0.75 | 1.60 | **1.63** | **1.04** | **2.57**\* |
| Southern or Eastern Africa | 39 | 5.91 | 43.39 | 1.93 | 0.99 | 3.74a | **2.25** | **1.04** | **4.90**\* |
| **Migration duration** | 655 | | | | | | | | |
| Living in Belgium for more than 10 years (or born) (ref.) | 237 | 36.24 | 31.22 | 1.00 | | | 1.00 | | |
| Not living in Belgium | 38 | 5.73 | 21.62 | 0.63 | 0.28 | 1.43 | 0.64 | 0.24 | 1.69 |
| Living in Belgium since <2 years | 150 | 22.86 | 32.89 | 1.08 | 0.70 | 1.68 | 1.03 | 0.58 | 1.81 |
| Living in Belgium for 2–10 years | 230 | 345.16 | 28.70 | 0.892 | 0.60 | 1.32 | 0.80 | 0.51 | 1.28 |
| **Legal status**[c] | 662 | | | | | | | | |
| Documented (ref.) | 519 | 78.45 | 27.94 | 1.00 | | | 1.00 | | |
| Probably undocumented | 143 | 21.55 | 39.16 | **1.65** | **1.12** | **2.43**\*\* | **1.83** | **1.12** | **2.99**\* |
| **Education** | 644 | | | | | | | | |
| Primary school or less (ref.) | 101 | 15.66 | 34.00 | 1.00 | | | 1.00 | | |
| Completed secondary | 311 | 48.36 | 33.12 | 0.96 | 0.60 | 1.54 | 1.03 | 0.60 | 1.78 |
| Continued education | 232 | 35.98 | 26.41 | 0.70 | 0.42 | 1.15 | 0.59 | 0.33 | 1.06a |
| **Employment** | 663 | | | | | | | | |
| (Self)employed | 315 | 47.61 | 29.43 | 1.00 | | | 1.00 | | |
| Unemployed | 189 | 28.47 | 32.80 | 1.17 | 0.79 | 1.72 | 0.82 | 0.51 | 1.31 |
| Full time student | 59 | 8.84 | 24.14 | 0.79 | 0.41 | 1.49 | 0.81 | 0.40 | 1.83 |
| Non-employed[d] | 100 | 15.08 | 30.00 | 1.17 | 0.72 | 1.89 | 0.85 | 0.47 | 1.56 |
| **Financial problems** | 624 | | | | | | | | |
| No (ref.) | 228 | 36.55 | 26.75 | 1.00 | | | 1.00 | | |
| Sometimes/most of the time | 396 | 63.45 | 33.16 | 1.37 | 0.95 | 1.96a | 1.28 | 0.85 | 1.92 |
| **Housing** | 635 | | | | | | | | |
| Stable (ref.) | 594 | 93.58 | 30.30 | 1.00 | | | 1.00 | | |
| Unstable | 41 | 6.42 | 40.00 | 1.56 | 0.81 | 2.98 | 1.45 | 0.65 | 3.26 |
| **Forced sex (lifetime)** | 646 | | | | | | | | |
| Never (ref.) | 602 | 93.25 | 29.40 | 1.00 | | | 1.00 | | |
| Ever | 44 | 6.75 | 47.73 | **2.24** | **1.21** | **4.16**\* | **2.18** | **1.03** | **4.57**\* |
| **Study setting** | 663 | | | | | | | | |
| Bar/party (ref.) | 342 | 51.63 | 29.15 | 1.00 | | | 1.00 | | |

*(Continued)*

**Table 1.** (Continued)

| Variable (N total)[a] | Total | | % | Un-adjusted | 95%-CI | | Adjusted | 95%-CI | |
|---|---|---|---|---|---|---|---|---|---|
| | N | % | eligible | OR | | | OR | | |
| Church | 160 | 24.19 | 29.81 | 1.03 | 0.69 | 1.56 | 1.16 | 0.70 | 1.91 |
| Public place | 75 | 11.32 | 32.00 | 1.12 | 0.66 | 1.93 | 0.99 | 0.54 | 1.83 |
| Info meeting | 34 | 5.18 | 40.00 | 1.59 | 0.77 | 3.28 | 2.08 | 0.92 | 4.69 |
| Other | 51 | 7.68 | 32.00 | 1.17 | 0.62 | 2.20 | 1.36 | 0.67 | 2.76 |

[*] $p<0.05$

[**] $p<0.01$

[***] $p<0.001$.

[a] n = 663 for the logistic regression analyses.

[b] ref. = reference category.

[c] operationalized by the proxy 'no health insurance'.

[d] including people who are retired, inactive due to sickness or disability, not allowed to work due to migration reasons, and housewives or men.

missing values on the seven eligibility criteria and were subsequently excluded from the sample for the bivariate and multivariable regression analyses (n = 663).

Graph 3 (Fig 3) presents the weighted percentages of people with SSA migrant background in Antwerp who met the PrEP eligibility criteria (see the corresponding S1 Table). In total, about 30.3% of them met at least one of the adapted criteria and thus were eligible for PrEP use. If we exclude the three specific criteria (7.1, 7.2 and 7.3) only 17.7% of the sample were eligible for PrEP. The most frequent criterion was 'having condomless sex with someone of SSA origin and being unaware of his/her HIV status' (16.4%), while a minority met the criterion 'diagnosed on a STI less than 6 months ago' (1.7%).

Among all HIV negative SSA migrants in our sample, 19.9% met only one criterion for PrEP eligibility, 7.1% met two criteria and 2.2% met three of them. The minority of SSA

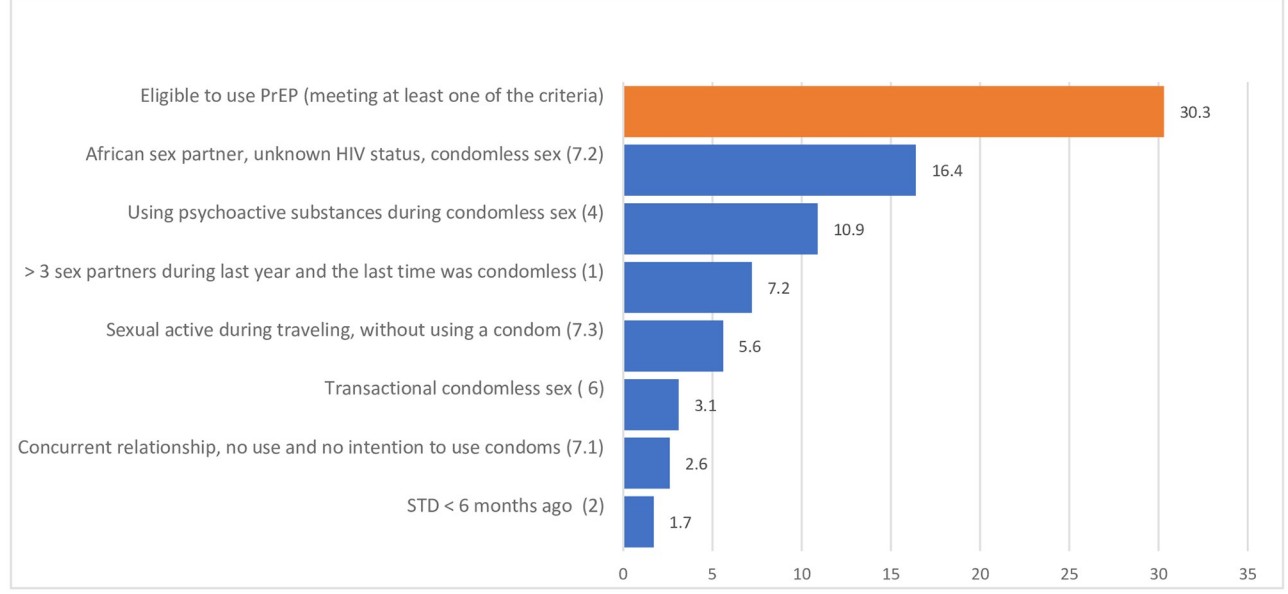

**Fig 3. Graph 3: Percentages of SSA migrants meeting the PrEP eligibility criteria[*].** [*]as presented in Box 2 (Fig 2) See also the corresponding S1 Table.

migrants meeting three or more criteria (3.4%; n = 23 out of n = 663) were a selective and vulnerable group: almost half had no health insurance, thus being probably undocumented, most of them lived less than two years in Belgium, were not in a relationship and experienced financial difficulties.

The bivariate results (Table 1) show that those who were eligible to use PrEP were significantly more likely to be male, MSM, single (versus in a relation and cohabiting) and reported having no health insurance than those who were ineligible. They were also more likely ever having experienced forced sex.

For the multivariable logistic regression analyses, there were no large differences between the results of Model 1, 2 and 3 when adding the socio-economic and forced sex variables. Therefore, only Model 3 is presented in Table 1 (adjusted ORs): Men, those without a relationship, those without health insurance, and who ever had experienced forced sex were more likely to be eligible for PrEP than women, those in a relationship and co-habiting, with health insurance, and those who had not experienced forced sex. Being MSM was no longer significantly associated with 'being eligible to PrEP use' when controlling for the other variables in the model. Educational level and region of origin became significantly associated to PrEP eligibility. Participants with only primary or no education and those from Central-Africa were more likely to meet one of the eligibility criteria when compared with those with vocational or university education (continued education) and people originating from Western Africa.

## Discussion

This study is the first to assess the eligibility to use PrEP among SSA migrants, residing in Belgium (Antwerp). Although only a few people with a SSA migration background use PrEP in Belgium, we have estimated that almost a third is eligible to use PrEP. To the best of our knowledge it is also the first study in Europe to quantify PrEP need among a mainly heterosexual population with migration background, in this case of SSA origin.

We found that 30.3% of the respondents were eligible to use PrEP. Those who were male, single, lower educated, without health insurance (thus likely of undocumented status), and those who had experienced forced sex were more likely to be eligible for PrEP use. Hence, a combination of factors at the individual, the interpersonal and the structural level may shape HIV risk and thus PrEP eligibility. This shows the significant role of multi-level social determinants of health, including migration, as found in other studies [10]. The high estimated proportion (roughly 30%) of SSA migrants who are eligible for PrEP is in contrast with the low number of people of SSA migration background actually taking PrEP in Belgium. This raises concerns about a potential large PrEP gap in this population.

The majority of those eligible for PrEP were meeting one or two eligibility criteria. However, a selective and vulnerable minority has three or more HIV risk factors: almost half had no health insurance, most of them lived less than two years in Belgium and experienced financial difficulties. Therefore, utmost attention should be paid to structural vulnerability such as having no health insurance or being undocumented. When people do not have any health insurance, they face a double risk. Due to their increased HIV risk, they have a higher need for PrEP, while at the same time access to and reimbursement of PrEP is more difficult, similar to the access of other HIV care services [1].

Adding interpersonal and network-level risk factors for PrEP elibility [12], such as having sex with people of African origin, concurrent relationships and sex during travelling after migration in Europe and Africa [1] increased the proportion of people with SSA migration background eligible for PrEP from 17.7% to 30.3%. This shows that it is important to consider target-group-specific factors accounting for HIV risk when assessing PrEP need. PrEP is the

most cost-effective [26] and has the highest impact on HIV prevalence [27] when those at highest risk of HIV exposure are identified and prioritized as potential PrEP users.

Our study highlights the role of several relevant predictors for PrEP eligibility. In what is to follow, we discuss the most important ones.

Respondents were more likely to be eligible for PrEP when they were male, single, had lower educational attainment, no health insurance, originated from Central, Southern and Eastern Africa, and had experienced forced sex. Several of these observations were in line with existing research on HIV vulnerability among SSA migrants [3, 4, 24]. Having no education or less than primary education and having no health insurance was related to a higher likelihood of being eligible for PrEP use, and SSA migrants who have financial problems had a higher chance of having condomless sex while taking psychoactive substances.

Our findings also highlight the role of intersecting vulnerabilities in shaping HIV risk and PrEP need: although the number of men having sex with men was limited in our sample (n = 13), more than half of them were eligible for PrEP use (n = 8). This reflects their high risk of HIV, and it is corroborated by the fact that the majority of the few people with SSA migrant background who already use PrEP are MSM, according to the Belgian HIV surveillance [28]. American research [29] reveals lower proportions of black MSM being eligible for PrEP based on the clinical criteria. They argue that this finding can be ascribed to the fact that the standardized clinical criteria are not sensitive to the risk factors among the black community, which is a concern that could also be raised about the Belgian guidelines. Our results also show that the subgroup of MSM reported more transactional condomless sex; were more often in a concurrent relationship having condomless sex; and reported lower condom use intentions compared to heterosexual men. As this sub-group belongs to both a sexual and an ethnic minority group confronted with multiple levels of stigma [30], they may require a targeted prevention approach.

## Limitations

Using an existing database comes with inherent limitations: we were unable to directly measure each criterion of the Belgian PrEP eligibility criteria (e.g. injecting drugs). Lack of these data might have resulted in an underestimation of the percentage of SSA migrants eligible to use PrEP. Self-reported and retrospective data may have led to underreporting of sensitive subjects (e.g. condomless sex) and may have been subject to recall bias. Data collection was limited to the city of Antwerp and therefore the results may have limited generalizability.

## Conclusions

In spite of these limitations, our findings enable us to draw a number of relevant recommendations for clinical practice and public health policy.

Based on our findings, we argue that the eligibility criteria in Belgium, and by extension in other countries, should include target-group specific risk factors, to increase the likelihood to detect unmet PrEP needs and enable these groups to start PrEP.

However, PrEP eligibility does not equal the willingness or intention to use PreP and as a consequence PrEP uptake [31]. We need to better understand the existing barriers for PrEP use within this population, and develop tailored ways to upscale PrEP use for those in need as part of a combination prevention strategy. Until now PrEP has been mostly been framed as a prevention method for MSM at high risk of HIV acquisition [11], while qualitative research has shown that among African communities it is often less known or perceived as a method for people with "promiscuous" sexual behavior [32].

Our results underscore the recommendations from research on black MSM in the US [12, 29]. To destigmatize PrEP and to reduce inequality in access to it, PrEP should be integrated into routine preventive health, primary care and in the Urgent Medical Care scheme. The latter ensures some free medical care among people without health insurances. However, these entitlements often go unrealized because of poor awareness of migrants' rights, fear of being reported to the immigration authorities and complex administrative procedures [33, 34]. Therefore, making PrEP easily available independent of health care coverage could be an option to increase access to PrEP care in Belgium (in analogy with HIV treatment), by both relieving the financial burden of PrEP and lowering the threshold for PrEP care for people in vulnerable situations, as illustrated by their financial problems and lack of health insurance. Discussing PrEP with all patients will reframe PrEP as sexual health promotion tool, irrespective of gender, sexual orientation, relationship status or ethnicity [12]. This way, unsensitive eligibility criteria and their implementation may be avoided, thus missing less potential PrEP candidates, above helping health care providers to raise the topic of HIV prevention without feeling uncomfortable or being stigmatizing.

In addition, our findings provide valuable insights into PrEP screening for specific subgroups within our target population, such as men with same sex behavior. The pre-established criteria seem not to be sufficient to capture the social and structural risk factors driving the HIV epidemic in this community. Accounting for the predictors at multiple levels, individual risk factors should be explored against the context of structural vulnerabilities. Screening questions aiming at revealing only individual sexual risk behaviors therefore may not pick up this vulnerability. In analogy with HIV testing [18, 35], more comprehensive guidance including culturally competent phrasing of screening questions should be developed.

Risk factors and the level of risk may also be dynamic over time [17, 32]. Likewise, PrEP need and eligibility can change related to life stages, residence status, relationship periods and socio-economic environment. This can be relevant in relation to mobility, where the likelihood of concurrent sexual relationships while traveling may be higher. Again, this highlights the importance of assessing such 'seasons of risk' during screening and providing PrEP alongside other appropriate interventions for a limited duration [32]. Time-limited use of PrEP and use related to specific situations and contexts may also have a positive impact on the accessibility and destigmatization of PrEP [32].

To conclude, the combination of a high proportion of people with SSA migration background who are eligible for PrEP use while almost none of them is currently using PrEP raises concerns about the effective implementation of PrEP among this population in Belgium. Future studies should inform evidence-based interventions to address the barriers to PrEP use at individual, community, and structural levels to achieve equitable access.

## Supporting information

**S1 Table. Percentages of SAM meeting the eligibility criteria (weighted data).**
(DOCX)

**S2 Table. Eligibility criteria associated to sociodemographic, economic, migration related factors and forced sex.**
(DOCX)

**S1 File. Interpretation of the results of the logistic regression analyses for the separate eligibility criteria.**
(DOCX)

**S2 File. TOGETHER project- database.**
(SAV)

# Acknowledgments

This study was a collaborative effort and is part of the Promise project 'Optimise PrEP to Maximise Impact'. The overall objective of this project is to learn how PrEP rollout can be optimized to result in maximum impact on HIV and sexual health. The data used stem from the TOGETHER project.

# Author Contributions

**Conceptualization:** Veerle Buffel, Caroline Masquillier, Thijs Reyniers, Ella Van Landeghem, Bea Vuylsteke.

**Data curation:** Veerle Buffel, Bea Vuylsteke, Christiana Nöstlinger.

**Formal analysis:** Veerle Buffel.

**Funding acquisition:** Thijs Reyniers, Edwin Wouters, Christiana Nöstlinger.

**Investigation:** Veerle Buffel, Caroline Masquillier, Thijs Reyniers, Ella Van Landeghem, Edwin Wouters, Bea Vuylsteke, Christiana Nöstlinger.

**Methodology:** Veerle Buffel, Bea Vuylsteke.

**Project administration:** Christiana Nöstlinger.

**Supervision:** Edwin Wouters, Bea Vuylsteke, Christiana Nöstlinger.

**Validation:** Veerle Buffel, Caroline Masquillier, Thijs Reyniers, Ella Van Landeghem, Edwin Wouters, Bea Vuylsteke, Christiana Nöstlinger.

**Writing – original draft:** Veerle Buffel.

**Writing – review & editing:** Veerle Buffel, Caroline Masquillier, Thijs Reyniers, Ella Van Landeghem, Edwin Wouters, Bea Vuylsteke, Christiana Nöstlinger.

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
