## [Decision Letter · Decision Letter 0]

3 Feb 2021

PONE-D-21-00363

Who falls between the cracks? Identifying eligiblity PrEP users among people with Sub-Saharan African migration background living in Antwerp, Belgium

PLOS ONE

Dear Dr. Buffel,

Thank you for submitting your manuscript to PLOS ONE. After careful consideration, we feel that it has merit but does not fully meet PLOS ONE’s publication criteria as it currently stands. Therefore, we invite you to submit a revised version of the manuscript that addresses the points raised during the review process.

From my own reading of the manuscript, I'm inclined to agree with the reviewer below that this manuscript is near ready for publication but could use a series of minor tweaks. For this reason, I have decided against waiting for a second reviewer's opinion and am sending on the below review. Please consider each item carefully and incorporate as you see fit.

We look forward to receiving your revised manuscript.

Kind regards,

Ethan Morgan

Academic Editor

PLOS ONE

Journal Requirements:

2. Please include additional information regarding the survey or questionnaire used in the study and ensure that you have provided sufficient details that others could replicate the analyses. For instance, if you developed a questionnaire as part of this study and it is not under a copyright more restrictive than CC-BY, please include a copy, in both the original language and English, as Supporting Information, or include a citation if it has been published previously.

3. In the Methods, please discuss whether and how the questionnaire was validated and/or pre-tested. If these did not occur, please provide the rationale for not doing so.

4. In statistical methods, please refer to any post-hoc corrections to correct for multiple comparisons during your statistical analyses. If these were not performed please justify the reasons. Please refer to our statistical reporting guidelines for assistance (https://journals.plos.org/plosone/s/submission-guidelines.#loc-statistical-reporting).

5.We note that the grant information you provided in the ‘Funding Information’ and ‘Financial Disclosure’ sections do not match.

6.We note that you have indicated that data from this study are available upon request. PLOS only allows data to be available upon request if there are legal or ethical restrictions on sharing data publicly. For more information on unacceptable data access restrictions, please see http://journals.plos.org/plosone/s/data-availability#loc-unacceptable-data-access-restrictions.

7. Please ensure that you refer to Figures 1, 2 and 3 in your text as, if accepted, production will need this reference to link the reader to the figure.

Reviewers' comments:

Reviewer's Responses to Questions

**Comments to the Author**

1. Is the manuscript technically sound, and do the data support the conclusions?

Reviewer #1: Yes

2. Has the statistical analysis been performed appropriately and rigorously? 

Reviewer #1: Yes

3. Have the authors made all data underlying the findings in their manuscript fully available?

Reviewer #1: Yes

4. Is the manuscript presented in an intelligible fashion and written in standard English?

Reviewer #1: Yes

5. Review Comments to the Author

Reviewer #1: This interesting article estimates the proportion of eligible PrEP users among SAM in Belgium. After MSM, SAM is indeed the second population who is at risk to acquire HIV and therefore, would benefit most from PrEP in Belgium. Results show that around 1/3rd of SAM is eligible for PrEP and those eligible are more likely to be male, MSM, single and not insured. In addition, they were also more likely ever having experienced forced sex. Identifying those men is crucial to further decrease HIV prevalence in Belgium.

Title: Please check the title: Identifying eligible PrEP users? (ok title in manuscript is correct)

Introduction:

- Line 51-52Do you have figures from 2019?

- Line 63-70: Line 63: you mentioned that 98% are MSM PrEP users, however line 70 you mention that only 1.1% of all applications for PrEP reimbursement came from SAM and in addition, 22/26 were MSM, so, in fact, only 4 heterosexual SAM requests were received? Are the 22 now accounted as SAM or as MSM using PrEP? Do you have any figures about the number of MSM in Belgium that is of SSA background?

Methods & Results:

- Your data used is from 2013-2014. Do you expect any changes over time?

- Box 2 : question 4. Psychoactive substances during sex. Indeed this is mentioned in the RIZIV guidelines, but I wonder if alcohol and cannabis is also included here. According to the definition of psychoactive substances it is, but then a high number of individuals could be included for PrEP… According to table S1, questions were asked separately. Only 14 individuals used drugs during sexual activity.

- Did you perform a sensitivity analysis if you used at least 2 criteria? (FYI: only 9.6% would be eligible…)

- Line 173: just out of interest. Do you have any idea how many men had sex with both genders?

- Line 177: retired is also considered to be unemployed. Were there many individuals that participated and were retired? Could that influence the results? It doesn’t seem correct to me to include those into unemployed.

- Line 185. Chi Square tests were used to determine whether the associations between these categorical variables … something is missing?

- Graph 3: the numbers after the eligibility criteria are confusing and they refer to box 2, so please mention it into the figure caption.

- Table1:

o Gender total is 663, same for relationship or region of origin.

o Male gender OR= 1.38 but 95% CI= 0.97-1.97. p<0.05, please check.

- In total 13 MSM were identified. Wouldn’t they already be informed about PrEP by their sexual networks? As mentioned above, are MSM from SSA origin now accounted as SAM or as MSM in Belgium (or both?)? Did the results change if you did not include those MSM into the analyses (n=13)?

- Line 276: this is partly methodology. Results of study setting should be included in table 1.

Discussion:

- Line 289: those without health insurance or likely to be eligible for PrEP. No health insurance and financial problems were correlated. What do you propose here? Without a health insurance PrEP is not available in Belgium, furthermore, in case they have health insurance, they still will need to pay 12€ for one bottle. In addition, visits to an ARC are not completely reimbursed and will therefore put an additional financial burden. Line 358 you mention that PrEP should be made available independent of health care coverage & should be integrated in primary care. I agree, however, will it then be completely free?

- Limitations:

o line 341, please check the sentence.

o I would also add that this data is from 7 years ago, therefore, behavioral changes during time could have taken place and your results could be outdated. For example, in the subsequent years we had an enormous immigration wave. Therefore, would it be of interest to relaunch the questionnaire?

o I’m still struggling with the psychotropic substance use during sex and your definition of use of alcohol or marihuana… However, strictly taken, it is correct.

- References: check ref 28 & 1. They are the same

6. PLOS authors have the option to publish the peer review history of their article (what does this mean?). If published, this will include your full peer review and any attached files.

Reviewer #1: No

---

## [Author Response · Author response to Decision Letter 0]

26 Mar 2021

Rebuttal letter

Dear Editor-in-Chief,

“Who falls between the cracks? Identifying eligible PrEP users among people with Sub-Saharan African migration background living in Antwerp, Belgium” 

Thank you for the opportunity to revise this article for possible publication in PLOS ONE. The amended manuscript reflects the authors’ efforts to address each of the comments and suggestions raised by the reviewers. 

The remainder of this letter gives specific details of how we addressed each comment, together with the line numbers (in the manuscript with track changes) upon which the relevant changes appear. 

As you indicated in your communication, the reviewer found much to like about the paper but also raises a series of minor tweaks and some suggestions for revising the paper. We also respond to the remaining editorial comments.

Reviewer #1: 

This reviewer valued the manuscript: “This interesting article estimates the proportion of eligible PrEP users among SAM in Belgium. After MSM, SAM is indeed the second population who is at risk to acquire HIV and therefore, would benefit most from PrEP in Belgium. Results show that around 1/3rd of SAM is eligible for PrEP and those eligible are more likely to be male, MSM, single and not insured. In addition, they were also more likely ever having experienced forced sex. Identifying those men is crucial to further decrease HIV prevalence in Belgium.”

Title

Comment 1: Please check the title: Identifying eligible PrEP users? (ok title in manuscript is correct)

We have checked the title in the system and the manuscript. 

Introduction:

Comment 2: Line 51-52: Do you have figures from 2019?

We have replaced the figures of 2018 by these of 2019 (which are the most recent numbers for Belgium). (line 51-52)

Sasse A, Deblonde J, De Rouck M, Montourcy M, Van Beckhoven D. Epidemiologie van aids en hiv-infectie in België toestand op 31 december 2019 [The epidemiology of AIDS and HIV infection in Belgian: the situation at the 31th of December 2019], Brussels: Sciensano; 2020.

Comment 3: Line 63-70, Line 63: You mentioned that 98% are MSM PrEP users, however line 70 you mention that only 1.1% of all applications for PrEP reimbursement came from SAM and in addition, 22/26 were MSM, so, in fact, only 4 heterosexual SAM requests were received? 

First of all, we have adjusted this paragraph to the most recent available figures (figures of 2019 instead of 2017). In 2019 only 1.4% of all applications for PrEP reimbursement stemmed from people with a SSA-migration background (equalling 25 people out of 1837 PrEP applicants with known nationality), of whom 18 were MSM). So only 7 heterosexual requests were received by people with nationality from a SSA country in 2019. (line 63-73)

Are the 22 now accounted as SAM or as MSM using PrEP? 

The report of Sciensano (the Belgian Institute of Public Health) presents the most recent numbers of PrEP-users based on the aggregated numbers of 11 of the 12 HIV reference centers in Belgian and Pharmanet (medication reimbursement data of all health insurances in Belgium). In their report the variables ‘nationality’ (Belgian, Sub-Sahara African countries and other) and ‘risk group’ (MSM, heterosexual, and other) are considered as two separate variables, which are not mutually exclusive: male SAM, who have sex with men are also counted as ‘MSM’; and MSM with a SSA-migration background are also counted as respondents with a SSA nationality. 

Sasse A, Deblonde J, De Rouck M, Montourcy M, Van Beckhoven D. Epidemiologie van aids en hiv-infectie in België toestand op 31 december 2019 [The epidemiology of AIDS and HIV infection in Belgian: the situation at the 31th of December 2019], Brussels: Sciensano; 2020.

Do you have any figures about the number of MSM in Belgium that is of SSA background?

No valid data about the number of MSM in Belgium are available, there are only some crude outdated estimations (Marcus, et al. 2013). Also accurate population size estimates of sub-Saharan African migrants are not available in Belgium. Officially 175,000 people who are born in a sub-Saharan African country are registered in Belgium (in 2014). This figure is an underestimation, as people of undocumented status, sub-Saharan African migrants who obtained Belgian nationality, and second and third generation migrants are not included in this number (Loos, et al. 2016). As a result, we also have no adequate Belgian figures about the number of MSM with a SSA-migration background. 

Marcus U, Hickson F, Weatherburn P, Schmidt AJ, Network E. Estimating the size of the MSM populations for 38 European countries by calculating the survey-surveillance discrepancies (SSD) between self-reported new HIV diagnoses from the European MSM internet survey (EMIS) and surveillance-reported HIV diagnoses among MSM in 2009. Bmc Public Health. 2013;13.

Loos J, Nostlinger C, Vuylsteke B, Deblonde J, Ndungu M, Kint I, et al. First HIV prevalence estimates of a representative sample of adult sub-Saharan African migrants in a European city. Results of a community-based, cross-sectional study in Antwerp, Belgium. Plos One. 2017;12(4). e0174677.https://doi.org/10.1371/journal.pone.0174677

Methods & Results:

Comment 4: Your data used is from 2013-2014. Do you expect any changes over time?

Unfortunately, no specific study has been conducted since 2014. We can only guess about changes over time. In Belgium, there is no regular behavioural monitoring on HIV/STI risk behaviours, as cohort data are only available for people living with HIV. As in other European countries, national surveillance data demonstrates a trend of an increasing ethnic diversity among the men having sex with men and newly reported heterosexually transmitted HIV infections, which partially may reflect changes in migration patterns.

Sasse A, Deblonde J, De Rouck M, Montourcy M, Van Beckhoven D. Epidemiologie van aids en hiv-infectie in België toestand op 31 december 2019 [The epidemiology of AIDS and HIV infection in Belgian: the situation at the 31th of December 2019], Brussels: Sciensano; 2020.

Comment 5: Box 2: question 4. Psychoactive substances during sex. Indeed this is mentioned in the RIZIV guidelines, but I wonder if alcohol and cannabis is also included here. According to the definition of psychoactive substances it is, but then a high number of individuals could be included for PrEP… 

This RIZIV criteria (‘MSM who use psychoactive substances during sexual activities’) is indeed quite broad, as they do not define ‘psychoactive substances’. We agree with the reviewer that this eligibility criteria is probably primarily referring to drug use during sexual activity and use of “chemsex”, as it is also formulated as a MSM-specific criteria. From previous research (Emis, 2019), we know that this is an important risk factor among this group. However, the Flemish expertise center for sexual health (‘Sensoa’) concretized this RIZIV criteria on their website by referring to drugs AND alcohol. We used a broad definition of psychoactive drugs including alcohol and drugs, as both alter people’s consciousness and this may influence sexual behavior. 

However, we are aware of the fact that by including alcohol and cannabis use in the operationalization of the criteria, the number of people with a SSA-migration background meeting this criteria has increased. Therefore, to avoid overestimations of the number of SAM eligible to use PrEP, we have added an additional condition to this criteria namely that the sexual activity under influence of drugs and/or alcohol was also condomless (‘Used drugs and/or alcohol during last sexual activity and this time was condomless’), which is not involved in the original RIZIV criteria. In this way, we have also adjusted this MSM specific eligibility measure to people with a SSA-migration background. 

In addition, following arguments have supported our decision to use this information about alcohol and marihuana use during sexual activity for the operationalization of this eligibility criteria:

• there is evidence that alcohol use just before or during sexual activity is related to poor decision-making and more sexual risk behavior (Baral, et al. 2011; Musinguzi, et al. 2015)

• in other countries (such as in Kenya) alcohol use just before or during sexual activities is used as a PrEP eligibility criteria (Wahome, et al. 2020)

• in the Together dataset, we only have about alcohol and marihuana use during last sexual activity, so otherwise, we could not include this item. Without this eligibility criteria, 25.9% (172/652) of the SAM were eligible instead of 29.6% (201/663). 

The EMIS Network. EMIS-2017 – The European Men-Who-Have-Sex-With-Men Internet Survey. Key findings from 50 countries. Stockholm: European Centre for Disease Prevention and Control; 2019.

Baral S, Adams D, Lebona J, Kaibe B, Letsie P, Tshehlo R, et al. A cross-sectional assessment of population demographics, HIV risks and human rights contexts among men who have sex with men in Lesotho. J Int Aids Soc. 2011;14.

Musinguzi G, Bastiaens H, Matovu JKB, Nuwaha F, Mujisha G, Kiguli J, et al. Barriers to Condom Use among High Risk Men Who Have Sex with Men in Uganda: A Qualitative Study. Plos One. 2015;10(7).

Wahome EW, Graham SM, Thiong'o AN, Mohamed K, Oduor T, Gichuru E, et al. PrEP uptake and adherence in relation to HIV-1 incidence among Kenyan men who have sex with men. EClinicalMedicine. 2020;26(100541).

Comments 6: According to table S1, questions were asked separately. Only 14 individuals used drugs during sexual activity.

Indeed, drugs seem to be less frequently used during sexual activity in comparison to alcohol among SAM. This is also one of the reasons why we included both the questions on alcohol and drug use during sexual activity. With only including ‘cannabis use during condomless sexual activity’ as criteria the percentage of SAM eligible to use PrEP was 26.6% compared to 29.6% (including both alcohol and cannabis use). 

Anecdotal evidence from prevention field workers also points into the direction that African communities are not or less engaged in party drug use typically associated with unprotected sex compared to Belgian MSM. Their community networks are different. But again, we lack evidence on these behaviors.

Comments 7: Did you perform a sensitivity analysis if you used at least 2 criteria? (FYI: only 9.6% would be eligible…)

If we would use ‘meeting at least 2 criteria’ instead of ‘at least one criteria’ to be eligible to use PrEP, 69 out of the 663 SAM (10.4%) would be eligible to use PrEP. However, this is not how ‘eligibility to use PrEP’ is defined in Belgium and other countries. The criteria need to be seen as risk factors of HIV, which we have applied to a population group who generally has a higher risk of acquiring HIV due to social vulnerabilities also related to the migration context, and reflected in their higher HIV prevalence and incidence rates in comparison to the general population. Therefore, refraining from conducting a sensitivity analysis is justified in our opinion.

Comment 8: Line 173: just out of interest. Do you have any idea how many men had sex with both genders?

In our dataset 3.1% (13 out of 421) of male SAM prefer a sexual partner of same sex or both. Among women with a SSA-background this is 3.8% (9 out of 239). However, we are not able to distinguish between SAM who preferred to have sex with both male and female partners and SAM who prefer to have sex with only one gender. 

Comment 9: Line 177: retired is also considered to be unemployed. Were there many individuals that participated and were retired? Could that influence the results? It doesn’t seem correct to me to include those into unemployed.

The Together dataset has no information about whether respondents have retired or not, we can only distinguish between: not working/not allowed to work, unemployed with or without benefits, employed with a permanent or a non-permanent contract, self-employed, student, and housewife/man or family caring. Our sample is also quite young with only 4 respondents who are 60 years old or older, and only 1 respondent is of legal retirement age (65 or above), so the number of people in retirement would be probable very low. 

However, we agree with the reviewer that ‘being unemployed’ is different to ‘being retired or non-employed’ (= inactive due to disability or sickness or who are not allowed to work because of immigration reasons). Unfortunately, this distinction was not made in the original questionnaire. Therefore, we have distinguished the non-employed from the unemployed in the revised manuscript. However the results do not differ a lot, as employment status is still not significantly related to ‘being eligible to use PrEP’ among SAM. (Line 183-184, and in Table 1).

Comment 10: Line 185. Chi Square tests were used to determine whether the associations between these categorical variables … something is missing?

We thank the reviewer for this remark and we adapted the text accordingly: ‘were significant’ was missing. (Line 201)

Comment 11: Graph 3: the numbers after the eligibility criteria are confusing and they refer to box 2, so please mention it into the figure caption.

We have made this clear in a note below the graph.

Table1:

Comment 12: Gender total is 663, same for relationship or region of origin.

We thank the reviewer for remarking this. 663 is the correct number (instead of 664). At the beginning of the sample selection one case was wrongly not excluded from the sample, which has led to this wrong total sample size (n). All figures have been adjusted to the correct sample size and the analyses were redone, but the results did not change substantially. 

Comment 13: Male gender OR= 1.38 but 95% CI= 0.97-1.97. p<0.05, please check.

We thank the reviewer for remarking this and have rechecked this result in Table 1. The revised table is adjusted to the correct results. 

Comment 14: In total 13 MSM were identified. Wouldn’t they already be informed about PrEP by their sexual networks? As mentioned above, are MSM from SSA origin now accounted as SAM or as MSM in Belgium (or both?)? Did the results change if you did not include those MSM into the analyses (n=13)?

In Belgium MSM from SSA origin are accounted as SAM and MSM. As requested by the reviewer, we have additionally estimated the unadjusted and adjusted odds ratio’s using a sample without the MSM. However, the results (see Table A in the 'response to reviewer' document) are still in line with the results of the analyses based on the total sample (inclusive the MSM) and we decided to keep the MSM included in the sample, because male SAM who have sex with men are meeting the inclusion criteria of our study and they highlight the importance of intersecting vulnerabilities in shaping PrEP need. 

TABLE A: Factors associated with eligibility to PrEP in bivariate and multivariate logistic analysis, based on the sample without the MSM (total n = 650) (weighted data) (see the 'Response to reviewers' file)

Comment 15: Line 276: this is partly methodology. Results of study setting should be included in table 1.

According to the reviewer’s comment, we have replaced the operationalization of the variable ‘study setting’ to the methodology part and the results of study setting are included in Table 1. (Line 192-194)

Discussion:

Comment 16: Line 289: those without health insurance or likely to be eligible for PrEP. No health insurance and financial problems were correlated. What do you propose here? Without a health insurance PrEP is not available in Belgium, furthermore, in case they have health insurance, they still will need to pay 12€ for one bottle. In addition, visits to an ARC are not completely reimbursed and will therefore put an additional financial burden. Line 358 you mention that PrEP should be made available independent of health care coverage & should be integrated in primary care. I agree, however, will it then be completely free?

In Belgium the coverage of the population by health insurance is ensured through the statutory compulsory system, which at 99% is virtually universal (Buffel & Nicaise, 2018). Despite almost universal insurance coverage, some groups systematically continue to fly under the radar, such as asylum seekers or undocumented migrants (AGII, 2014). Although Belgium has a special scheme to ensure some free medical care in case of emergency and essential healthcare, these entitlements often go unrealised because of poor awareness of the rights, fear of being reported to the immigration authorities and complex administrative procedures (Derluyn et al., 2011). This urgent medical care scheme is also formally not including preventive health care, such as PrEP care. 

We have knowledge of a few organizations who can provide PrEP for free for certain populations often without a health insurance and with intersecting vulnerabilities (such as initiatives for male and transgender sex workers) - but this is often highly dependable on the goodwill of the physicians and social organizations involved (for example through fundraising activities). Our argument to make PrEP available independent of health care coverage is aimed at both relieving the financial burden of PrEP, as to lowering the threshold for PrEP (and HIV related) care for people in vulnerable situations - illustrated by their financial problems and lack of health insurance.

In addition, even when people have a health insurance the out-of-pocket payment of PrEP is still relatively high. There are several initiatives aim to limit the total amount of personal contributions that a patient actually has to pay (for example a ‘right to increased health insurance reimbursement’ statute, which provides preferential tariffs for persons of specific social status or on the basis of their income), but also these systems have a relatively high non-take-up rate especially among very vulnerable groups (Buffel & Nicaise, 2018). Moreover, PrEP care is indeed more than providing PrEP medication alone, which also led to indirect costs like transport, etc.

In conclusion, this adds the following knowledge gap: we do not know the total financial burden of PrEP for PrEP patients in vulnerable contexts, since all of the above does not apply to the general white male MSM PrEP user, further research on this is necessary. 

See additional explanation in the manuscript at lines 369-377

Agentschap integratie en inburgering (AGII) Kruispunt Migratie en Integratie vzw, Infofiche ‘Wanneer hebben vreemdelingen recht op een ziekte verzekering?’ [When are foreigners entitled to sickness insurance?’], Expertisecentrum voor Vlaanderen-Brussel (Expertise Centre of Flanders-Brussels), 2014, at: http://www.agii.be/sites/default/files/bestanden/documenten/document_infofiche_ziekteverzekering_vreemdelingen.pdf

Buffel, V & Nicaise I (2018) “ESPN Thematic Report on Inequalities in access to healthcare” Belgium, European Social Policy Network. 

Derluyn, I., Lorant, V., Dauvrin, M., Coune, I. and Verrept, H., ‘Naar een interculturele gezondheidszorg: Aanbevelingen van de ETHEALTH-groep voor een gelijkwaardige gezondheid en gezondheidszorg voor migranten en etnische minderheden’ [Towards intercultural healthcare: Recommendations from the ETHEALTH group for an equal health and healthcare for migrants and ethnic minorities], 2011, at: https://www.unia.be/files/Z_ARCHIEF/2012_12_16_eindrapport_NL.pdf

Limitations:

Comment 17: line 341, please check the sentence.

Modifications can be found on line 351.

(Data collection was limited to the city of Antwerp and may therefore the results have limited generalizability � Data collection was limited to the city of Antwerp and therefore the results may have limited generalizability).

Comment 18: I would also add that this data is from 7 years ago, therefore, behavioral changes during time could have taken place and your results could be outdated. For example, in the subsequent years we had an enormous immigration wave. Therefore, would it be of interest to relaunch the questionnaire?

We thank the reviewer for raising this comment. Indeed, relaunching a bio-behavioural seroprevalence study should be a research and policy priority to reflect changes both in migration context (in this case including not only migrants from SSA but from diverse backgrounds) and in the HIV prevention landscape, i.e. introduction of PrEP, a stronger emphasis on U=U messages.

Used as a second generation behavioural surveillance tool, coupled with HIV testing (i.e. eventually shorter versions with the most important indicator) it could guide new prevention interventions on which target populations, which settings to prioritise. 

Comment 19: I’m still struggling with the psychotropic substance use during sex and your definition of use of alcohol or marihuana… However, strictly taken, it is correct.

Please see our reply on comment 5.

References: 

Comment 20: check ref 28 & 1. They are the same

We thank the reviewer for this remark and excluded ref 28. 

Editorial comments:

Comment 1: Please ensure that your manuscript meets PLOS ONE's style requirements, including those for file naming. 

 We have revised the manuscript in line with the style requirements. 

Comment 2: Please include additional information regarding the survey or questionnaire used in the study and ensure that you have provided sufficient details that others could replicate the analyses. For instance, if you developed a questionnaire as part of this study and it is not under a copyright more restrictive than CC-BY, please include a copy, in both the original language and English, as Supporting Information, or include a citation if it has been published previously.

In this study, we performed a secondary analysis of data from a cross-sectional bio-behavioral survey of the TOGETHER study. As mentioned in the manuscript details about this survey (validation of the questionnaire, recruitment, data collection and weighting procedures) can be found in the protocol paper and a first descriptive paper, i.e. the seroprevalence study. (line 95-96 and 102-104)

Loos J, Vuylsteke B, Manirankunda L, Deblonde J, Kint I, Namanya F, et al. TOGETHER Project to Increase Understanding of the HIV Epidemic Among Sub-Saharan African Migrants: Protocol of Community-Based Participatory Mixed-Method Studies. JMIR Res Protoc. 2016;5(1):e48.

Loos J, Nostlinger C, Vuylsteke B, Deblonde J, Ndungu M, Kint I, et al. First HIV prevalence estimates of a representative sample of adult sub-Saharan African migrants in a European city. Results of a community-based, cross-sectional study in Antwerp, Belgium. Plos One. 2017;12(4). e0174677.https://doi.org/10.1371/journal.pone.0174677

Comment 3: In the Methods, please discuss whether and how the questionnaire was validated and/or pre-tested. If these did not occur, please provide the rationale for not doing so.

The details about the questionnaire can be found in the protocol paper, where we refer to in the manuscript (line 95-96 and 102-104).

We gladly provide information about the validation and development of the TOGETHER questionnaire, as described in the protocol paper: “The structured electronic questionnaire was developed based on the findings of formative study 2 [i.e. a multiple case study aiming to identify risk and vulnerability factors for HIV acquisition], consultation of available questionnaires from comparable studies, and input from the community researchers and the Community Advisory Board. It was refined after cognitive piloting with 12 participants and the pilot sessions.”

Loos J, Vuylsteke B, Manirankunda L, Deblonde J, Kint I, Namanya F, et al. TOGETHER Project to Increase Understanding of the HIV Epidemic Among Sub-Saharan African Migrants: Protocol of Community-Based Participatory Mixed-Method Studies. JMIR Res Protoc. 2016;5(1):e48.

Comment 4: In statistical methods, please refer to any post-hoc corrections to correct for multiple comparisons during your statistical analyses. If these were not performed please justify the reasons. Please refer to our statistical reporting guidelines for assistance (https://journals.plos.org/plosone/s/submission-guidelines.#loc-statistical-reporting).

In confirmatory studies, which may lead to a change in clinical practice or approval of a new treatment, it is more important to guard against the possibility of false-positive results due to multiple comparisons (Feise, 2002, Althouse, 2016). When it comes to exploratory studies or post-hoc analysis of existing data (like this study), though, a strict adjustment for multiple comparisons is less critical, as long as the manuscript contains a clear statement acknowledging that and declaring that a subsequent study with preplanned hypotheses should be conducted to confirm the observed association. We followed the approach proposed by Athouse (2016): ‘describe what was done in the study, report effect sizes, confidence intervals and p values and let readers use their own judgment about the relative weigh of the conclusions’. 

Althouse A, D, (2016) Adjust for Multiple Comparisons? It’s Not That Simple, The Annals of Thoracic Surgery, Volume 101, Issue 5, 1644 – 1645. 

Feise, R.J. Do multiple outcome measures require p-value adjustment?. BMC Med Res Methodol 2, 8 (2002). https://doi.org/10.1186/1471-2288-2-8

Comment 5: We note that the grant information you provided in the ‘Funding Information’ and ‘Financial Disclosure’ sections do not match.

The data used in this study stem from the TOGETHER project, which was funded by the Scientific Fund for Research on AIDS, managed by the King Baudouin Foundation, Belgium.

However, the current study is part of the Promise project, which is funded by FWO-SBO (Flemish Scientific Research – Strategic Basic Research). This is the reason why we provided two funding sources in the section ‘funding information’ and the ‘Acknowledgments’ in the manuscript, but we only have a grant number of the current FWO-SBO project ‘Promise’.

Comment 6: When you resubmit, please ensure that you provide the correct grant numbers for the awards you received for your study in the ‘Funding Information’ section.

We have removed the funding source of the Together data and only included FWO-SBO as funding source of the current project ‘Promise’, whereof we have a grant number. (Please see also our answer above on comment 5)

We note that you have indicated that data from this study are available upon request. PLOS only allows data to be available upon request if there are legal or ethical restrictions on sharing data publicly. For more information on unacceptable data access restrictions, please see http://journals.plos.org/plosone/s/data-availability#loc-unacceptable-data-access-restrictions.

We uploaded the minimal anonymized data set (as supplementary information, S3 File), necessary to replicate our findings. In fact, this data set is already available at PloS One in another article published on this data. 

https://journals.plos.org/plosone/article?id=10.1371/journal.pone.0174677#sec020

(line 219)

Loos J, Nostlinger C, Vuylsteke B, Deblonde J, Ndungu M, Kint I, et al. First HIV prevalence estimates of a representative sample of adult sub-Saharan African migrants in a European city. Results of a community-based, cross-sectional study in Antwerp, Belgium. Plos One. 2017;12(4). e0174677.https://doi.org/10.1371/journal.pone.0174677

Comment 7: Please ensure that you refer to Figures 1, 2 and 3 in your text as, if accepted, production will need this reference to link the reader to the figure.

In the revised manuscript we have referred correctly to our figures: Box 1 and 2 are now labelled as Fig 1 and Fig 2 and Graph 3 as Fig 3.

---

## [Decision Letter · Decision Letter 1]

1 Jul 2021

PONE-D-21-00363R1

Who falls between the cracks? Identifying eligiblity PrEP users among people with Sub-Saharan African migration background living in Antwerp, Belgium

PLOS ONE

Dear Dr. Buffel,

Thank you for submitting your manuscript to PLOS ONE. After careful consideration, we feel that it has merit but does not fully meet PLOS ONE’s publication criteria as it currently stands. Therefore, we invite you to submit a revised version of the manuscript that addresses the points raised during the review process.

One of the peer reviewers has identified a few minor areas that need additional clarification/editing. I'm confident that once these are addressed, we can proceed with the publication process. While they are minor, they will add clarity to your manuscript. 

We look forward to receiving your revised manuscript.

Kind regards,

Anthony J. Santella, DrPH, MPH, MCHES

Academic Editor

PLOS ONE

Journal Requirements:

Reviewers' comments:

Reviewer's Responses to Questions

**Comments to the Author**

1. If the authors have adequately addressed your comments raised in a previous round of review and you feel that this manuscript is now acceptable for publication, you may indicate that here to bypass the “Comments to the Author” section, enter your conflict of interest statement in the “Confidential to Editor” section, and submit your "Accept" recommendation.

Reviewer #1: All comments have been addressed

Reviewer #2: (No Response)

2. Is the manuscript technically sound, and do the data support the conclusions?

Reviewer #1: Yes

Reviewer #2: Yes

3. Has the statistical analysis been performed appropriately and rigorously? 

Reviewer #1: Yes

Reviewer #2: Yes

4. Have the authors made all data underlying the findings in their manuscript fully available?

Reviewer #1: Yes

Reviewer #2: Yes

5. Is the manuscript presented in an intelligible fashion and written in standard English?

Reviewer #1: Yes

Reviewer #2: Yes

6. Review Comments to the Author

Reviewer #1: Thank you for the clarification. I have no additional comments. The publication can be published. Best Wishes

Reviewer #2: Thank you for giving me the opportunity to review this revised manuscript. I thought it was well written and the subject is interesting. I do have some questions and point of feedback that will hopefully help to further improve manuscript.

Major:

- Line 163-165 & results 272-287: While I do understand why this extra analysis is interesting, I found that it was a distraction from the main question the authors wanted to answer in this paper and it made the Results section a bit confusing and long. I would suggest to either take these analyses out of the paper or move the results to the Supplements to keep the paper concise.

- Line 196-198: I did not fully understand what the authors meant by control variable study setting. Do you mean this variable was controlled for in multivariable analysis? Did you expect differences between different study setting? A suggestion would be to use multilevel analysis to account for this.

Minor:

- Line 50-52: Could you add some more context here on the number of new HIV infections in 2019 and what proportion of new HIV infections in 2019 was among heterosexual individuals?

- Line 66: Is it a box or a Figure? It now says Fig 1. Box 1.

- Line 101: what do you mean by two-stage time location sampling?

- Line 212-214: Were all variables assessed in univariable analysis included in the multivariable model? Were all variables kept in the model when going to the next Model (i.e. were all variables from Model 1, still in Model 2)?

- Line 229: do you mean multivariable regression analysis instead of multiple?

- Line 250: please use multivariable here unless you added multiple outcome variables to your model.

7. PLOS authors have the option to publish the peer review history of their article (what does this mean?). If published, this will include your full peer review and any attached files.

Reviewer #1: No

Reviewer #2: No

---

## [Author Response · Author response to Decision Letter 1]

13 Jul 2021

Rebuttal letter

13th of July 2021

Dear Editor-in-Chief,

“Who falls between the cracks? Identifying eligible PrEP users among people with Sub-Saharan African migration background living in Antwerp, Belgium” 

Thank you for the opportunity to revise this article for publication in PLOS ONE. The amended manuscript reflects the authors’ efforts to address the additional minor comments and suggestions raised by the reviewer. 

The remainder of this letter gives specific details of how we addressed each comment, together with the line numbers (in the manuscript with track changes) upon which the relevant changes appear. 

As you indicated in your communication, the first reviewer is satisfied with our previous adaptations and the second reviewer has identified a few minor areas that need additional clarification and editing. We also respond to the remaining editorial comments.

Yours sincerely,

Dr. Veerle Buffel

Reviewer #1: 

Thank you for the clarification. I have no additional comments. The publication can be published. Best Wishes

Reviewer #2: 

Thank you for giving me the opportunity to review this revised manuscript. I thought it was well written and the subject is interesting. I do have some questions and point of feedback that will hopefully help to further improve manuscript.

Major:

Comment 1: Line 163-165 & results 272-287: While I do understand why this extra analysis is interesting, I found that it was a distraction from the main question the authors wanted to answer in this paper and it made the Results section a bit confusing and long. I would suggest to either take these analyses out of the paper or move the results to the Supplements to keep the paper concise.

Reply:

We agree with the remark of the reviewer, as this was also the reason why the results of these extra analyses were already only displayed as supplementary material (S2 Table. Eligibility criteria associated to sociodemographic, economic, migration related factors and forced sex). According to the suggestion of the reviewer, we have replaced also the interpretation of these results to the Supplements (S3 Supporting information. Interpretation of the results of the logistic regression analyses for the separate eligibility criteria). (Lines 231-234)

In the manuscript:

We employed logistic regression models using a stepwise approach: to socio-demographic and migration related factors (Model 1) we added socio-economic factors (Model 2) and in the last step we also included the variable forced sex (Model 3). To get a better understanding of each HIV risk factor, the same analyses were also done with the individual eligibility criteria as dependent dichotomous variables. These results and their interpretation can be found respectively in S2 Table and S3 Supporting information. 

Comment 2: Line 196-198: I did not fully understand what the authors meant by control variable study setting. Do you mean this variable was controlled for in multivariable analysis? Did you expect differences between different study setting? A suggestion would be to use multilevel analysis to account for this.

Reply:

In general, we did not expect differences in ‘eligibility to PrEP use’ across the study settings, as the respondents were randomly selected among the people present at the study setting at time of recruitment. The study settings were chosen during formative research as places were people originating from SSA often get together. Only for some specific risk factors, we could expect some differences: for example, people recruited at a bar or party could have a higher likelihood to meet the risk factor ‘using psychoactive substances during condomless sex’ than people recruited at the church (this was also confirmed in the bivariate analyses).

Indeed, we also considered a multilevel analysis, but there was no significant variance at the higher level (study setting) and there were no higher level variables we want to estimate on ‘PrEP eligibility’. Also the results of the bivariate analyses (Chi2-tests) showed no significant differences between the study settings in terms of ‘eligibility to PrEP use’. In addition, by controlling for study setting through the use of dummies, we were able to assess the differences between the respondents in terms of ‘eligibility to use PrEP’ according to the study setting where they were recruited. Because of these reasons we prefer a one-level logistic regression analysis with the inclusion of the variable ‘study setting’ over a two-level analysis with study setting as the higher level to account for the clustering of and potential similarities between participants recruited from the same study setting. 

In the revised manuscript we have added some information about the two-stage time location sampling and the selection of study settings (sites where SSA migrants often get together). This has made it easier to explain clearly the control variable ‘study setting’. (lines 106-110 and 206-210)

In the manuscript:

The TOGETHER data are the most recent behavioral data available for this sub-population in Belgium. The study used a two-stage time-location sampling (TLS) to obtain a venue-based sample of n=744 adult sub-Saharan African migrants in Antwerp (1). A TLS takes advantage of the fact that some hard-to-reach populations tend to gather at certain types of sites/clusters at certain times. A list of these sites was prepared in a formative study and formed the sampling frame: at the first level, clusters (or sites) were randomly selected with a probability proportional-to-size and at the second level, a fixed number of study participants were randomly selected from each cluster. 

…

The study sites where study participants were selected were categorized in five types of settings and this is included as control variable: bars/parties of African organization, churches, public place (park, street, square), events and meetings of African organizations, and other (e.g. shop, hair salon, library, asylum center)(1).

Minor:

Comment 3: Line 50-52: Could you add some more context here on the number of new HIV infections in 2019 and what proportion of new HIV infections in 2019 was among heterosexual individuals?

Reply:

Based on a recent report of Sciensano (the public health institution of Belgium), we have added some more information about the new HIV infections in 2019 and the proportion of heterosexual transmissions among these new HIV cases in Belgium. (Lines 50-55)

In the manuscript:

Introduction

People with a Sub-Saharan Africa (SSA) migration background are the second largest group affected by HIV in Belgium (1). In 2019, a total of 923 new HIV diagnoses were identified in Belgium, among them 349 were through heterosexual transmission (51% of the cases with known transmission type) (2). People with SSA migration background constituted 48% of all new HIV cases with a heterosexual transmission mode in 2019, 67% among them were women (2). …

2. Sasse A, Deblonde J, De Rouck M, Montourcy M, Van Beckhoven D. Epidemiologie van aids en hiv-infectie in België toestand op 31 december 2019 [The epidemiology of AIDS and HIV infection in Belgian: the situation at the 31th of December 2019], Brussels: Sciensano; 2020.

Comment 4: Line 66: Is it a box or a Figure? It now says Fig 1. Box 1.

Reply: This figure is a box, but according to the submission system of the Journal we could only include this box as ‘figure’ 

Comment 5: Line 101: what do you mean by two-stage time location sampling?

Reply: 

A 2-stage time location sampling (TLS) takes advantage of the fact that some hard-to-reach populations tend to congregate at certain types of locations. A list of these locations (settings) is prepared in a formative study and forms the sampling frame from which a 2-stage cluster probability sample was selected. At the first level of sampling clusters (or sites) were randomly selected with a probability proportional-to-size and at the second level, from each cluster a random selection of a fixed number of study participants was done. In the revised manuscript we have briefly explained the TLS and for more detailed information we refer to the protocol paper of the Together project. (lines 106-110)

In the manuscript:

The TOGETHER data are the most recent behavioral data available for this sub-population in Belgium. The study used a two-stage time-location sampling (TLS) to obtain a venue-based sample of n=744 adult Sub-Saharan African migrants in Antwerp (1). A TLS takes advantage of the fact that some hard-to-reach populations tend to gather at certain types of sites/clusters at certain times. A list of these sites was prepared in a formative study and formed the sampling frame: at the first level, clusters (or sites) were randomly selected with a probability proportional-to-size and at the second level, a fixed number of study participants were randomly selected from each cluster. 

All individuals socializing in a given setting at the time of the study visit (available attendance data) were eligible for inclusion in the survey if they met the following criteria: (1) self-identified sub-Saharan African migrant; (2) age 18 years or above; (3) accepting to answer the questionnaire; (4) accepting to provide an oral fluid sample; and (5) providing written informed consent. Recruitment, data collection and weighting procedures to adjust for unequal selection probability are described elsewhere (1, 16).

Comment 6: Line 212-214: Were all variables assessed in univariable analysis included in the multivariable model? Were all variables kept in the model when going to the next Model (i.e. were all variables from Model 1, still in Model 2)?

Reply: Yes, we have used a stepwise procedure. This means that we have added variables to the next models, while the other variables of the previous model remained included. 

Comment 7: Line 229: do you mean multivariable regression analysis instead of multiple?

Reply: Yes, indeed, thanks for remarking this. It is changed in the revised manuscript. 

Comment 8: Line 250: please use multivariable here unless you added multiple outcome variables to your model.

Reply: We have changed it to ‘multivariable’ because the analysis has multiple independent variables but only one outcome variable. 

Journal Requirements:

Reply: We have revised the reference list according to the PLOSONE guidelines.

---

## [Decision Letter · Decision Letter 2]

4 Aug 2021

Who falls between the cracks? Identifying eligiblity PrEP users among people with Sub-Saharan African migration background living in Antwerp, Belgium

PONE-D-21-00363R2

Dear Dr. Buffel,

We’re pleased to inform you that your manuscript has been judged scientifically suitable for publication and will be formally accepted for publication once it meets all outstanding technical requirements.

Kind regards,

Anthony J. Santella, DrPH, MPH, MCHES

Academic Editor

PLOS ONE

Additional Editor Comments (optional):

Reviewers' comments:

Reviewer's Responses to Questions

**Comments to the Author**

1. If the authors have adequately addressed your comments raised in a previous round of review and you feel that this manuscript is now acceptable for publication, you may indicate that here to bypass the “Comments to the Author” section, enter your conflict of interest statement in the “Confidential to Editor” section, and submit your "Accept" recommendation.

Reviewer #1: All comments have been addressed

Reviewer #2: All comments have been addressed

2. Is the manuscript technically sound, and do the data support the conclusions?

Reviewer #1: Yes

Reviewer #2: Yes

3. Has the statistical analysis been performed appropriately and rigorously? 

Reviewer #1: Yes

Reviewer #2: Yes

4. Have the authors made all data underlying the findings in their manuscript fully available?

Reviewer #1: Yes

Reviewer #2: Yes

5. Is the manuscript presented in an intelligible fashion and written in standard English?

Reviewer #1: Yes

Reviewer #2: Yes

6. Review Comments to the Author

Reviewer #1: I did not have any additional comments since the previous revision round. Well done and good luck...

Reviewer #2: (No Response)

7. PLOS authors have the option to publish the peer review history of their article (what does this mean?). If published, this will include your full peer review and any attached files.

Reviewer #1: No

Reviewer #2: No

---

## [Editor Report · Acceptance letter]

9 Aug 2021

PONE-D-21-00363R2 

Who falls between the cracks? Identifying eligible PrEP users among people with Sub-Saharan African migration background living in Antwerp, Belgium 

Dear Dr. Buffel:

I'm pleased to inform you that your manuscript has been deemed suitable for publication in PLOS ONE. Congratulations! Your manuscript is now with our production department. 

Kind regards, 

on behalf of

Dr. Anthony J. Santella 

Academic Editor

PLOS ONE